# A Novel Tandem-Tag Purification Strategy for Challenging Disordered Proteins

**DOI:** 10.3390/biom12111566

**Published:** 2022-10-26

**Authors:** Attila Mészáros, Kevin Muwonge, Steven Janvier, Junaid Ahmed, Peter Tompa

**Affiliations:** 1VIB-VUB Center for Structural Biology, Vlaams Instituut voor Biotechnologie (VIB), 1050 Brussels, Belgium; 2Structural Biology Brussels (SBB), Vrije Universiteit Brussel (VUB), 1050 Brussels, Belgium; 3Research Centre for Natural Sciences (RCNS), Institute of Enzymology, ELKH, 1117 Budapest, Hungary

**Keywords:** intrinsically disordered proteins (IDPs), protein purification, affinity chromatography, Tau, androgen receptor (AF1)

## Abstract

Intrinsically disordered proteins (IDPs) lack well-defined 3D structures and can only be described as ensembles of different conformations. This high degree of flexibility allows them to interact promiscuously and makes them capable of fulfilling unique and versatile regulatory roles in cellular processes. These functional benefits make IDPs widespread in nature, existing in every living organism from bacteria and fungi to plants and animals. Due to their open and exposed structural state, IDPs are much more prone to proteolytic degradation than their globular counterparts. Therefore, the purification of recombinant IDPs requires extra care and caution, such as maintaining low temperature throughout the purification, the use of protease inhibitor cocktails and fast workflow. Even so, in the case of long IDP targets, the appearance of truncated by-products often seems unavoidable. The separation of these unwanted proteins can be very challenging due to their similarity to the parent target protein. Here, we describe a tandem-tag purification method that offers a remedy to this problem. It contains only common affinity-chromatography steps (HisTrap and Heparin) to ensure low cost, easy access and scaling-up for possible industrial use. The effectiveness of the method is demonstrated with four examples, Tau-441 and two of its fragments and the transactivation domain (AF1) of androgen receptor.

## 1. Introduction

The production of recombinant proteins is a crucial technique in both academic research and industrial applications [1]. In industry, such as the pharma sector, the use of biopharmaceuticals is becoming the dominant trend. In the last few years, close to 100 new biopharmaceuticals, the majority being recombinant proteins, have entered the market [2]. In molecular biology, recombinant protein purification is a vital technique for a broad range of applications, such as structural characterization by X-ray crystallography, nuclear magnetic resonance (NMR), small-angle X-ray scattering (SAXS) and cryo-electron microscopy (cryo-EM) [3,4,5]. Most of these require large quantities of protein with high quality, although cryo-EM is less stringent on sample requirement following its compatibility with protein purified from native samples [6]. Besides protein structure technologies, other in vitro biochemical studies and molecular biology applications also require protein of good quality and reasonable quantity. Therefore, significant effort is directed towards developing new and improved purification approaches [7,8,9].

Recombinant protein purification can be classified by the host organism in which the expression is performed [10]. Each system has its advantages and disadvantages, and the host of choice is often motivated by the downstream application of the purified protein [11,12]. Bacterial expression systems, for example, are well-established, easy to handle and relatively cheap [13]. However, they provide proteins without typical post-translational modifications (PTMs), which might be critical for the native, functional state of eukaryotic target proteins [14]. Furthermore, removing bacterial endotoxin from the sample can be cumbersome, compromising biotechnological applications [15]. Bacterial expression can also fail to produce soluble proteins, as many of them tend to form inclusion bodies (IBs) [16,17]. Eukaryotic systems, on the other hand, appear to be superior in producing soluble and active eukaryotic proteins [18,19,20,21]. However, their final yield tends to be lower, and the production process is more labor intensive, requiring special media and equipment, which may significantly increase the total cost of production. Another variation on the theme is the production of proteins in so-called cell-free systems. In this case, instead of using a host organism, an in vitro mixture is reconstituted for protein expression [22]. These systems are relatively costly and are not suited for high-level protein expression, even though they allow fast production and incorporation of special amino acids [23].

The most common host organism for recombinant protein production is *Escherichia coli* (*E. coli*) [24]. As an expression system, *E. coli* is easy to handle, cheap, has a high growth rate and usually produces large quantities of the desired recombinant protein [25]. As mentioned earlier, its major disadvantage is the lack of PTMs, which can result in improper folding and/or IB formation of expressed eukaryotic proteins. Furthermore, bacterial codon usage, which differs from that of eukaryotes, may also be a limiting factor. This is more evident in the case of human recombinant proteins expressed in bacteria. All in all, this system is still widely used due to its many advantages, as intense research is also being conducted to overcome its limitations [11,25,26,27,28]. For example, the introduction of solubility tag(s) or co-expression of molecular chaperones can significantly improve solubility of expressed proteins [29,30,31,32]. Codon bias can also be overcome via codon optimization, or by using special strains that contain transfer RNAs (tRNAs) at levels typical of eukaryotes [24,33]. Genetic modification of *E. coli* strains even allows the production of glycosylated antibodies [13,34,35].

Intrinsically disordered proteins (IDPs) are proteins that lack well-defined 3D structures [36]. Since their discovery, there has been a boom in studies highlighting their important roles in crucial cellular processes [37]. Due to their lack of well-defined 3D structures, IDPs are much more susceptible for proteolytic degradation [13,38]. For this reason, extra precaution needs to be taken during their purification, such as applying protease inhibitors, keeping the temperature low or optimizing for a very fast workflow. Despite all precautions, however, degradation still occurs most of the time. One approach to overcome the challenges of degradation is the development of tandem-tag based methods, applying different affinity tags on both termini of the protein of interest [39,40,41,42,43]. In such instances, however, only one tag is normally removed by targeted proteolytic cleavage, to ensure that the remaining tag can be used in subsequent applications such as pull-down or western-blot experiments [40,41,42]. In cases where both tags must be removed, this is usually achieved via two different proteases [43]. The other drawback of some tag combinations reported in the literature is that the necessary column and the elution reagents are either expensive or not easily available (e.g., FLAG-tag^®^ or Twin-Strep-tag^®^) [39,41,42,44]. In addition, there are other plasmid constructs that contain two affinity tags for consecutive affinity steps to ensure higher purity, although in most cases the two tags are located at the same terminus [7].

Here, we describe a novel tandem-tag based method for IDP purification, in which both tags can be removed simultaneously. We demonstrate the versatility of the method on a few selected IDP examples with different charge properties.

## 2. Materials and Methods

### 2.1. Generation of pSUMO Plasmid

To generate the pSUMO plasmid for our method, we modified an existing one (pHYRSF53, a gift from Hideo Iwai, Addgene plasmid # 64696; http://n2t.net/addgene:64696, accessed on 25 September 2022; RRID: Addgene_64696) [45]. pHYRSF53 contains an N-terminal 6xHis-tag followed by a SUMO tag, and it was used as a backbone. First, a C-terminal DNA-binding domain (DBD) was added to the pHYRSF53 plasmid using the HiFi DNA Assembly Cloning kit (New England Biolabs (NEB), Ipswich, MA, USA), following the instructions of the manufacturer. The plasmid used as a template to generate the DBD fragment of androgen receptor was created in-house. The fragment was generated by polymerase chain reaction (PCR) using Q5 High-Fidelity DNA polymerase (NEB, Ipswich, MA, USA), following the manufacturer’s instructions. After the successful insertion of the DBD fragment into pHYRSF53, we inserted a multiple cloning site (MCS) flanked by two TEV cleavage sites in between the two affinity tags. The MCS-TEV-site fragment was ordered as a single-stranded DNA fragment from Eurofins Genomics (Ebersberg, Germany). Transformants in NEB 5-alpha chemically competent bacterial cells were then selected using Luria Bertani (LB)-agar containing 50 µg/mL Kanamycin antibiotic (Duchefa Biochemie, Haarlem, The Netherlands). Single colonies were picked and grown overnight at 37 °C in LB broth media (Duchefa Biochemie, Haarlem, The Netherlands) supplemented with 50 µg/mL Kanamycin antibiotic (LB-Kanamycin). Plasmid DNA was then isolated from the liquid cultures using the MN-NucleoSpin Plasmid QuickPure kit (Fisher Scientific, Merelbeke, Belgium), and plasmid DNA sequences were verified by Sanger sequencing (Microsynth, Balgach, Switzerland).

### 2.2. Generation of pSUMO Expression Constructs

#### 2.2.1. Generating pSUMO Constructs by HiFi Cloning (pSUMO-AF1, pSUMO-Tau-441)

We generated a pSUMO-AF1 plasmid for recombinant protein expression by inserting a DNA fragment coding for the activation function 1 (AF1) domain of Androgen receptor into the MCS of pSUMO plasmid by HiFi cloning. The AF1-coding DNA fragments were generated with Q5 High-Fidelity DNA polymerase, using a plasmid housing a coding sequence of full-length AF1 as a template (The plasmid containing AF1-coding sequence was also generated in-house). Similarly, the pSUMO-Tau-441 plasmid for recombinant protein expression was generated by inserting the full-length Tau-441-coding sequence into the pSUMO plasmid using the HiFi DNA Assembly Cloning kit.

In both cases, successful transformants in NEB 5-alpha bacterial cells were selected on LB-Kanamycin agar plates, and single colonies were then cultured in LB-Kanamycin liquid media for plasmid extraction. Plasmid DNA was isolated using MN-NucleoSpin Plasmid QuickPure kit, and plasmid DNA sequences were verified by Sanger sequencing.

#### 2.2.2. Generating pSUMO Constructs by Site-Directed Mutagenesis (pSUMO-Tau-NTMT, pSUMO-Tau-MTBR and pSUMO-AF1 (Only N-tag))

The pSUMO-Tau-NTMT plasmid was generated by site-directed mutagenesis of the pSUMO-Tau-441 construct via deletion of Tau-441′s short C-terminal tail. The pSUMO-Tau-MTBR construct was then generated by further mutagenesis of pSUMO-Tau-NTMT plasmid, via deletion of the flexible N-terminal region of Tau-441. The pSUMO-AF1(only N-tag) construct was created by inserting a stop codon (TAG) at the end of the AF1 coding sequence, such that only the N-terminal 6xHis-SUMO-tag would be translated as a fusion to AF1 recombinant protein, without the C-terminal DBD-affinity tag.

In all cases, mutagenesis was performed using the Q5 High-Fidelity DNA polymerase kit following the manufacturer’s instructions. After the mutagenesis PCR, nascent non-methylated DNA strands harboring respective DNA modifications were enriched by KLD (Kinase, Ligase and Dpn1) treatment (NEB, Ipswich, MA, USA). Successful transformants in NEB 5-alpha bacterial cells were selected on LB-Kanamycin agar plates, and single colonies were then cultured in LB-Kanamycin liquid media for plasmid extraction. Plasmid DNA was isolated using the MN-NucleoSpin Plasmid QuickPure kit, according to the manufacturer’s instructions. Plasmid DNA sequences of pSUMO-Tau-NTMT, pSUMO-Tau-MTBR and pSUMO-AF1(only N-tag) were confirmed by Sanger sequencing before expression of recombinant proteins.

### 2.3. Expression and Purification of pSUMO Constructs

#### 2.3.1. Expression and Purification of pSUMO-AF1(Only N-tag)

As the pSUMO-AF1(only N-tag) construct was not codon optimized for *E. coli* expression, it was expressed in *E. coli* Rosetta 2 cells (Invitrogen, Waltham, MA, USA) to enhance expression of such a eukaryotic protein in a bacterial system. Cells were cultured at 37 °C in Terrific Broth (TB) that was prepared in-house and supplemented with 50 µg/mL of Kanamycin and 25 µg/mL of Chloramphenicol antibiotics (Duchefa Biochemie, Haarlem, The Netherlands). Recombinant protein expression was induced with 1 mM isopropyl β-D-1-thiogalactopyranoside (IPTG) (Sigma-Aldrich, St. Louis, USA) at an optical density (OD_600_) of 1.2. Cells were then cultured for another 5 h at 30 °C, harvested by centrifugation (Avanti JXN-26, Beckman Coulter, CA, USA) at 5000 revolutions per minute (rpm) for 15 min, and bacterial pellets were stored at −80 °C awaiting protein purification.

To purify pSUMO-AF1, 1 L of bacterial pellet was resuspended in 75 mL of Lysis buffer, composed of 50 mM Tris (Sigma-Aldrich, St. Louis, USA), 250 mM NaCl (Sigma-Aldrich, St. Louis, USA), 25 mM Imidazole (Merck, Darmstadt, Germany), 10% Glycerol (VWR, Ohio, USA), 0.1% Triton X-100 (Sigma-Aldrich, St. Louis, USA), 0.5 mM tris(2-carboxyethyl) phosphine (TCEP) (VWR, Ohio, USA), 1 mM phenyl-methyl-sulfonyl fluoride (PMSF) (Sigma-Aldrich, St. Louis, USA), 2 tablets of Roche cOmplete protease inhibitor cocktail (Merck, Darmstadt, Germany), pH 8.0) using a glass homogenizer (Carl Roth, Karlsruhe, Germany). Cells were then lysed by sonication (5 s on/5 s off, 60% amplitude, 5 min), and the resulting bacterial lysate centrifuged for 1 h at 40,000× *g* (4 °C) in presence of 1 mg/mL DNase I (Sigma-Aldrich, St. Louis, USA). The lysate supernatant was filtered using a 0.45 µm syringe filter (Sarstedt, Nümbrecht, Germany) and every step was carefully performed on ice.

The filtered lysate was loaded onto a HisTrap-HP-5 mL column (Cytiva, Uppsala, Sweden) using an AKTA™ Pure chromatography system (GE Healthcare, Uppsala, Sweden) in a cooling cabinet. The HisTrap loading buffer (A) was 50 mM Tris, 250 mM NaCl, 25 mM imidazole, 0.5 mM TCEP, pH 8.0. After sample loading, a washing step was applied with buffer A to wash away unbound contaminants. For elution, a gradient was applied using an elution buffer (B) composed of 50 mM Tris, 250 mM NaCl, 500 mM Imidazole, 0.5 mM TCEP, pH 8.0. Elution fractions were analyzed by sodium dodecyl-sulfate polyacrylamide gel electrophoresis (SDS-PAGE), and selected fractions were pooled together in a sterile 50-mL Falcon tube (Greiner Bio-One, Vilvoorde, Belgium).

To the pooled fractions, 0.1 mg/mL of TEV protease (From a 10 mg/mL stock that was expressed and purified in-house) was added to cleave the N-terminal 6xHis-SUMO-tag. The digested protein mixture was dialyzed against buffer A in a cold-room maintained at 4 °C, and then taken through a Reverse HisTrap purification using HisTrap loading (A) and elution (B) buffers. The HisTrap flow-through was dialyzed against 50 mM Tris, 150 mM NaCl, 0.5 mM TCEP, pH 8.0, and concentrated by ultrafiltration (VivaSpin, 10 kDa MWCO, Sartorius, Stonehouse, UK) in preparation for size exclusion chromatography (SEC), which was performed using a Superdex XK 16/100 (200 pg) column (Cytiva, Uppsala, Sweden). After SEC, elution fractions were analyzed by SDS-PAGE, and selected fractions were pooled and concentrated as before. The protein concentration of the final product was determined by measuring the absorbance at 280 nm, using NanoDropTMOne (ThermoFisher Scientific, Waltham, MA, USA). The yield was calculated to 1 L of bacterial culture (0.8 mg protein/L).

#### 2.3.2. Expression and Purification of pSUMO-AF1

The pSUMO-AF1 construct was also not codon optimized for *E. coli* expression, and hence its recombinant expression was performed in *E. coli* Rosetta 2 cells. The cells were cultured at 37 °C in TB containing 50 µg/mL of Kanamycin and 25 µg/mL of Chloramphenicol antibiotics. Protein expression was induced with 1 mM IPTG at OD_600_ = 1.2, and cells were cultured for another 5 h at 30 °C. Cells were then harvested by centrifugation at 5000 rpm for 15 min, and the bacterial pellets stored at −80 °C awaiting protein purification.

Just like pSUMO-AF1(only N-tag) purification, a 1 L pSUMO-AF1 bacterial pellet was resuspended in 75 mL of a Lysis buffer with identical composition. Resuspended cells were then lysed, centrifuged, and the lysate supernatant filtered as previously described.

The filtered lysate was loaded onto a HisTrap-HP-5 mL column equilibrated with HisTrap loading buffer (A1) composed of 50 mM Tris, 250 mM NaCl, 25 mM imidazole, 0.5 mM TCEP, pH 8.0. After sample loading, a high-salt ATP-wash buffer (A2) composed of 50 mM Tris, 500 mM NaCl, 500 mM KCl (VWR, Leuven, Belgium), 25 mM Imidazole, 10 mM ATP (Merck, Darmstadt, Germany), 20 mM MgCl_2_ (Sigma-Aldrich, St. Louis, USA), 0.5 mM TCEP, pH 8.0, was applied to the column to wash away bacterial chaperone contaminants that co-purify with AF1. Bound proteins were eluted from the column by applying a gradient of elution buffer (B) composed of 50 mM Tris, 250 mM NaCl, 500 mM imidazole, 0.5 mM TCEP, pH 8.0. Elution fractions were analyzed by SDS-PAGE and selected fractions were pooled together in a sterile 50-mL Falcon tube.

Pooled fractions were dialyzed against Heparin loading buffer A (50 mM Tris, 5 mM TCEP, pH 8.0) and loaded onto a Heparin-HP-5 mL chromatography column (Cytiva, Uppsala, Sweden) equilibrated with the same buffer. Bound proteins were washed with Heparin buffer A and eluted using a gradient of Heparin Buffer B (50 mM Tris, 1 M NaCl, 5 mM TCEP, pH 8.0). After SDS-PAGE analysis of collected samples, selected fractions were pooled and mixed with 0.4 mg/mL TEV protease to simultaneously cleave both affinity tags. The protein mixture was then dialyzed overnight at 4 °C against HisTrap Buffer A1 and taken through a Reverse HisTrap purification using the same HisTrap buffers outlined above (except the buffer A2). The HisTrap flow-through was then dialyzed against Heparin Buffer A, concentrated by ultrafiltration (VivaSpin 10 kDa MWCO) and taken through a final Reverse Heparin purification step using buffers and Heparin chromatography column outlined above. The Reverse Heparin flow-through was further concentrated by ultrafiltration as before, and the protein concentration was determined by measuring the absorbance at 280 nm, using NanoDropTMOne (ThermoFisher Scientific, Waltham, MA, USA). The yield for 1 L bacterial culture was then calculated (4.6 mg protein/L).

#### 2.3.3. Expression and Purification of pSUMO-Tau-441

The pSUMO-Tau-441 construct was codon optimized for *E. coli* expression; therefore, the recombinant protein was expressed in *E. coli* BL21 Star™ (DE3) cells (NEB, Ipswich, MA, USA). Cells were cultured at 37 °C in TB supplemented with 50 µg/mL Kanamycin antibiotic. Protein expression was induced with 1 mM IPTG at OD_600_ = 1.2, then cells were cultured for another 5 h at 30 °C. Bacterial cells were then harvested by centrifugation at 5000 rpm for 15 min and the bacterial pellets stored at −80 °C awaiting protein purification.

To purify pSUMO-Tau-441, bacterial pellet from one liter of medium was resuspended in 75 mL of Lysis buffer (50 mM HEPES (Sigma-Aldrich, St. Louis, MO, USA), 0.2 mM MgCl_2_, 10% Glycerol, 0.1% Triton X-100, 5 mM TCEP, 1 mM PMSF, 2 tablets of Roche cOmplete protease inhibitor cocktail, pH 7.2) using a glass homogenizer. Cells were then lysed by sonication (5 s on/5 s off, 60% amplitude, 5 min), and bacterial lysate centrifuged for 1 h at 40,000× *g* (4 °C), in the presence of 1 mg/mL DNase I (Sigma-Aldrich, St. Louis, MO, USA). The lysate supernatant was filtered using a 0.45 µm syringe filter and every step was carefully performed on ice.

The filtered lysate was then loaded onto a Heparin-HP-5 mL chromatographic column equilibrated with Heparin buffer A (50 mM HEPES, 0.5 mM TCEP, pH 7.2). Bound proteins were washed with Heparin buffer A and eluted using a gradient of Heparin Buffer B (50 mM Tris, 1 M NaCl, 0.5 mM TCEP, pH 7.2). Elution fractions were analyzed by SDS-PAGE and selected fractions were pooled together in a 50-mL Falcon tube. After adding 25 mM imidazole to the pooled fractions, the protein solution was loaded onto a HisTrap-Hp-5 mL column equilibrated with HisTrap buffer A (50 mM HEPES, 250 mM NaCl, 25 mM imidazole, 0.5 mM TCEP, pH 7.2). Bound proteins were washed with HisTrap buffer A and eluted using a gradient of HisTrap Buffer B (50 mM HEPES, 250 mM NaCl, 500 mM imidazole, 0.5 mM TCEP, pH 7.2). After analyzing the fractions on SDS-PAGE, selected fractions were pooled together in a sterile 50-mL Falcon tube. From this point on, the purification followed the workflow employed for pSUMO-AF1, while using buffers outlined for pSUMO-Tau-441. The final yield was normalized to one liter of bacterial culture (2.5 mg protein/L).

#### 2.3.4. Expression and Purification of pSUMO-Tau-NTMT

The pSUMO-Tau-NTMT recombinant protein was also expressed to high levels in *E. coli* BL21 Star™ (DE3) cells, as it was generated from the pSUMO-Tau-441 construct codon optimized for bacterial expression. Cells were cultured at 37 °C in TB supplemented with 50 µg/mL of Kanamycin antibiotic. Protein expression was induced with 1 mM IPTG at OD_600_ = 1.2, and cells were cultured for another 5 h at 30 °C. Cells were then harvested by centrifugation at 5000 rpm for 15 min, and bacterial pellets were stored at −80 °C awaiting protein purification.

To purify pSUMO-Tau-NTMT, 1 L of bacterial pellet was resuspended in 50 mL of Lysis buffer (50 mM HEPES, 250 mM NaCl, 0.2 mM MgCl_2_, 10% Glycerol, 0.1% Triton X-100, 0.5 mM TCEP, 1 mM PMSF, 1 tablet of Roche cOmplete protease inhibitor cocktail, pH 7.2) using a glass homogenizer. Cells were then lysed by sonication (5 s on/5 s off, 60% amplitude, 5 min) and the bacterial lysate was centrifuged for 1 h at 40,000× *g* (4 °C) in the presence of 1 mg/mL DNase I. The lysate supernatant was filtered using a 0.45 µm syringe filter, and every step was carefully performed on ice.

The purification of pSUMO-Tau-NTMT followed the workflow described for pSUMO-AF1 with minor differences highlighted below. HisTrap loading buffer (A) was composed of 50 mM HEPES, 250 mM NaCl, 25 mM imidazole, 0.5 mM TCEP, pH 7.2, whereas HisTrap elution buffer (B) was composed of 50 mM HEPES, 250 mM NaCl, 500 mM imidazole, 0.5 mM TCEP, pH 7.2. Similarly, the Heparin loading buffer (A) was HEPES-based at pH 7.2, with 250 mM NaCl and 1 mM TCEP. The Heparin elution buffer (B) contained additional salt (1 M NaCl) for gradient elution of proteins bound to the Heparin chromatography column. After the final purification step, the protein concentration of purified pSUMO-Tau-NTMT was determined by measuring absorbance at 205 nm (instead of 280 nm), considering that the Tau-NTMT polypeptide sequence lacked any Tryptophan residues, the major contributors to intrinsic fluorescence at 280 nm. The final yield was normalized to one liter of bacterial culture (1.8 mg protein/L).

#### 2.3.5. Expression and Purification of pSUMO-Tau-MTBR

The shorter pSUMO-Tau-MTBR construct was expressed and purified in the same way as pSUMO-Tau-NTMT, with minor modifications. Despite attaining high-level expression of Tau-MTBR at 30 °C for 5 h when induced with 0.5 mM IPTG, a large portion of the expressed protein localized to bacterial inclusion bodies. This was probably because bacterial chaperones could not keep up with the translation machinery rapidly producing large amounts of aggregation-prone Tau-MTBR, resulting in the sequestration into inclusion bodies [40]. To overcome this, slow expression of pSUMO-Tau-MTBR was adopted where protein expression was induced by 0.5 mM IPTG, and bacteria further cultured overnight at 16 °C. This approach dramatically improved on the yield recovered in the soluble fraction of the bacterial lysate (Figure 6). In addition, unlike Tau-NTMT, purified Tau-MTBR was dialyzed and concentrated using 3 kDa MWCO dialysis membrane (Serva, Heidelberg, Germany) and VivaSpin ultrafiltration columns (Sartorius, Stonehouse, UK), respectively, due to the relatively smaller size of the cleaved final product (15.8 kDa). The final yield was normalized to one liter of bacterial culture (2.1 mg protein/L).

### 2.4. LC-MS/MS Analysis

SDS-PAGE bands of interest were subjected to in-gel digestion with trypsin and ProteaseMax™ surfactant, both obtained from Promega (Maddison, WI, USA), following the manufacturer’s instructions. Processed samples were then snap-frozen and stored at −80 °C awaiting further analysis.

LC-MS/MS analysis of the tryptic digests was performed by means of a Q-Exactive™ Focus Hybrid Orbitrap mass spectrometer equipped with a Thermo Scientific™ Vanquish™ ultra-high performance liquid chromatography system (Thermo Fisher Scientific, Waltham, MA, USA). Five microliters of the tryptic digests were injected and chromatographically separated by means of a 35-min linear gradient of 2–45% mobile phase B (mobile phase A: 0.1% formic acid, mobile phase B: 0.1% formic acid in acetonitrile) on an Acquity UPLC^®^ CSH C18 column (2.1 × 150 mm, 1.7 µm) from Waters (Milford, MA, USA). The flow rate and column temperature were 0.3 mL/min and 45 °C, respectively. The Q-Exactive Focus, operating in data dependent acquisition mode (DDA), was set to perform a mass spectrometry (MS) scan (R= 70 000 at 200 m/z, AGC target 3.0 e6) from 375 to 1500 m/z, followed by HCD MS^2^ spectra (R= 17 500 at 200 m/z, NCE =27%, AGC target = 1.0e5, max ion time = 50 ms) on the three most abundant precursors (quadrupole isolation width 1.4 m/z).

Data treatment and data analysis were performed by PEAKS studio 10.6 (Bioinformatics Solutions Inc., Waterloo, Canada). *De novo* sequencing and database searches (full Swiss-Prot database, downloaded 12 September 2022) were performed with a 10-ppm precursor mass tolerance, a 0.02 Da fragment tolerance and an FDR <0.1% on the peptide level. Oxidations of methionine were set as a variable modification, while the carbamidomethylation of cysteines was included as a fixed modification. Only fully tryptic peptides and a maximum of three trypsin mis-cleavages were allowed. Database searches were performed with and without the AA sequences of the respective DNA constructs present.

## 3. Results

### 3.1. Generation of pSUMO Plasmid with Tandem-Tags

Generally, many expression plasmids contain one affinity tag fused to either the N- or C-terminus of the target protein for subsequent purification by affinity chromatography (Figure 1A). This is usually sufficient for many proteins. However, as we already mentioned in the Introduction, IDPs are prone to proteolytic degradation due to their lack of a stable structure [13,38]. This degradation leads to truncated products that still carry one of the affinity-tags; hence these unwanted products usually co-purify with the target protein. Therefore, extra purification steps such as ion-exchange (IEX) or size exclusion (SEC) chromatography are necessary to get rid of these degradation fragments. This extra step, however, can be challenging because of the high similarity and/or small size difference between the target protein and its truncated versions. As mentioned above, a second affinity step can isolate the intact protein from a heterogeneous mixture [39,40,41,42,43]. We generated a plasmid for bacterial expression (bearing the T7 promoter) containing two different affinity-tags at the N- and C-terminus of the target protein, respectively (Figure 1B). In between the tags and the gene of interest, TEV cleavage sites were cloned into the plasmid, to facilitate the simultaneous removal of both tags. On the N-terminus, a SUMO tag was inserted to ensure a high expression level and increased solubility of the target protein [46]. The SUMO tag itself does not participate in affinity purification, hence we combined it with a 6xHis-tag, the most commonly available tag for HisTrap purification. On the C-terminus, the DNA-binding domain (DBD) of androgen receptor was inserted. Heparin mimics DNA, therefore, DBDs can specifically bind to heparin columns [47]. However, heparin is not specific enough to enable a one-step purification, as it can also non-specifically bind other proteins. It can be applied, however, if there is another affinity step in the workflow [48]. These heparin columns have also been reported to behave as cation exchangers [49].

Upon partial proteolytic degradation, one of the two termini is truncated, which leads to the malfunction of the tag at the affected termini. By applying two subsequent affinity steps (HisTrap and Heparin), we could ensure that only the intact protein is isolated from a bacterial cell lysate. Following successful isolation, both tags can be simultaneously removed in a single TEV protease cleavage step (Figure 2A). As the TEV protease also carries a fused 6xHis-tag at its N-terminus, the free 6xHis-SUMO-tag, uncleaved target proteins and the protease can be simultaneously removed by a reverse HisTrap purification step (Figure 2B). Reverse Heparin purification acts as a final polishing step. Moreover, since the flow-through is collected in these last steps, non-specifically binding contaminants are also going to re-bind to the columns, further improving the overall purity of the final product.

### 3.2. Purification of pSUMO-AF1 and pSUMO-AF1(Only N-tag): Comparison of the Two Methods

Androgen receptor is a transcription factor consisting of three domains: a globular DNA binding domain (DBD) sandwiched between a C-terminal ligand binding domain and a disordered N-terminal domain (NTD) [51]. The NTD contains an activation function domain (AF1) that is responsible for the recruitment of important cofactors, hence it has been studied extensively [52]. The existing purification strategy of AF1 recombinant protein is based on a single HisTrap via a 6xHis-tag followed by a SEC polishing step [53].

We compared our method with others applied for purifying the same protein. The purification of pSUMO-AF1(only N-tag) that contains only the N-terminal 6xHis-tag followed by the SUMO-tag relies on only one affinity purification step (HisTrap). This approach suffers from a high level of degradation by-products, which are very difficult to separate due to the resolution limit of the SEC columns (Appendix A). Furthermore, as only a small portion of the fractions contains the intact protein, the yield is usually very low. In contrast, the purification of pSUMO-AF1 containing a DBD as an additional C-terminal tag for a second affinity purification (Heparin) resulted in recovery of intact protein, which was verified by MS (Appendix A). Interestingly, after the first HisTrap, we observed two peaks. Analyzing the contents of these peak fractions by SDS-PAGE revealed that both peaks contained the desired product. However, one peak was almost entirely clean, containing only the intact protein with both tags, but the second peak also contained truncated degradation fragments (Appendix A). As a precaution, these fractions were pooled separately for downstream purification steps (HisTrap pool I and II). The second affinity purification step (Heparin) did not improve on the purity of the first pool (HisTrap pool I), but it significantly improved the purity of HisTrap pool II (Appendix A). Nonetheless, we pooled interesting elution fractions separately for further purification steps (Heparin pool I and II). In the next step, both tags were removed by targeted proteolytic cleavage using TEV protease. The protease, uncleaved products and 6xHis-SUMO-tag were simultaneously removed in the reverse HisTrap step, and the fully cleaved product was recovered in the flow-through. As a final polishing step, another Heparin affinity purification (reverse Heparin) was performed. This served to remove the cleaved DBD-tag that bound to the Heparin column as the final product was collected in the flow-through (Figure 3). The final cleaved product was verified by MS (Appendix A). The faint impurity in the final product was AF1 with uncleaved DBD that dimerized with fully cleaved AF1 and eluted together in the flow-through. This can be avoided by adding more reducing agent to the sample just before the reverse Heparin chromatography step as demonstrated later in the purification of Tau-MTBR.

It is important to note that, after performing all the purification steps, there was no difference in the purity between reverse Heparin pool I and II (Figure 3). This shows that the purification method worked equally well on the clean (HisTrap pool I) and dirty (HisTrap pool II) fractions of the first HisTrap step, demonstrating its separation capability and robustness.

### 3.3. Purification of pSUMO-Tau Constructs

#### 3.3.1. Purification of pSUMO-Tau-441

Microtubule-associated protein Tau has been implicated in various neurodegenerative diseases, such as Alzheimer’s disease (AD) and frontotemporal dementia (FTD) [54]. It has multiple isoforms in the central nervous system, with the longest human isoform consisting of 441 amino acid residues, named Tau-441 (it is also known as htau40 and Tau-2N4R). Full-length Tau-441 is made up of a long flexible N-terminal region, a microtubule-binding region (MTBR) and a short C-terminal tail. Tau-441 is highly dynamic and behaves as an IDP in solution. Therefore, its structure has only been studied using NMR spectroscopy [54,55]. Recently, Tau-441 has been shown to undergo liquid-liquid phase separation (LLPS), which seems to play a key role in both its physiological functions and pathological aggregate formation [56,57,58]. Generally, LLPS and NMR studies require proteins of high purity and yield, which can only be satisfactorily produced in a bacterial expression system. Due to its intrinsically disordered nature, Tau-441 can withstand high temperature, thus the most common Tau purification methods in the literature include a boiling step on top of affinity chromatographic separation [57,58]. Nonetheless, the bacterial expression and purification of soluble and intact Tau-441, as well as its domain constructs, is not straightforward due to their open and exposed structural state; they are highly prone to proteolytic degradation. Moreover, they contain aggregation-prone motifs that can lead to solubility issues and sequestration into bacterial inclusion bodies [40].

To overcome these challenges, we applied our tandem-tag purification method to purify Tau-441 and two of its domains. Application of tandem tags can provide a remedy for truncated degradation products, while the N-terminal SUMO-tag also enhances solubility of the protein.

In the case of Tau-441, we changed the approach and decided to start the purification with Heparin chromatographic separation, as this method—unlike HisTrap—is not sensitive to high concentrations of reducing agents. It has been recently proposed that Tau molecules form higher-order oligomers via disulfide bonding of their cysteine residues, which can later lead to aggregate formation [59]. Therefore, we opted to use increased amounts of reducing agent in the lysis buffer (5 mM TCEP) to counter disulfide-bond formation among Tau monomers. As expected, Heparin affinity purification alone was not sufficient to produce a high purity sample, but it was able to enrich the target protein in separate fractions with significantly decreased levels of contamination (Appendix A). Elution fractions that contained our protein of interest were pooled together for the next affinity purification step. Another advantage of this approach was that the elution buffer from the previous purification step (Heparin buffer B) was more compatible with the loading buffer of the next step (HisTrap buffer A). We could thus proceed without a buffer exchange of the Heparin pool sample (Appendix A). The intact Tau-441 protein containing both tags was verified by MS (Appendix A). In the next step, both tags were removed by TEV digestion and the method was continued the same way as described above for pSUMO-AF1. Interestingly, during the last reverse Heparin step, we isolated the final cleaved product in the elution fractions, rather than in the flow-through (Figure 4). This can be explained by the isoelectric point (pI) of the target protein (pI 7.85), coupled with the cation exchange behavior of Heparin columns. Successful cleavage of both affinity tags in the final product was also confirmed by MS analysis (Appendix A).

#### 3.3.2. Purification of pSUMO-Tau-NTMT

Biomolecular condensation of Tau has emerged as a crucial process in both its physiological microtubule-associated functions and pathological aggregation leading to neurodegeneration [56,57]. The N-terminal half of Tau-441 coupled to the microtubule-binding domain (Tau-NTMT; residues 1 to 372 of Tau-441) have been identified as the minimal construct driving its LLPS via intramolecular and intermolecular electrostatic interactions [56]. The purification of this construct from bacteria presents similar challenges as full-length Tau-441, considering that it only lacks a very short C-terminal tail, with its flexible N-terminal region and the aggregation-prone microtubule-binding region still present. We have attempted to purify this recombinant protein using our tandem-tag purification strategy following a similar workflow to that of pSUMO-Tau-441. Surprisingly, not much of the desired protein was recoverable when starting with Heparin affinity purification instead of the HisTrap purification (data not shown). This could be indicative of the lower binding capacity of the Heparin-HP-5 mL chromatographic column when compared to that of a HisTrap-HP-5 mL column, and not the binding affinity of the DBD-tag versus the 6xHis-tag to their respective affinity columns. It is for this reason that the purification strategy where the HisTrap preceded Heparin chromatographic separation, followed by affinity-tag cleavage using TEV protease and eventually separating the cleaved tags from purified Tau molecules, was employed for these two Tau domain constructs.

Considering that most of the expressed recombinant protein was in the soluble fraction of bacterial whole-cell lysate, pSUMO-Tau-NTMT was the most dominant protein bound to the HisTrap column in the first chromatographic step (Appendix A). Therefore, all elution fractions containing pSUMO-Tau-NTMT were pooled, dialyzed against Heparin buffer A, and loaded onto the Heparin-HP-5 mL column. Here, pSUMO-Tau-NTMT was favored to bind to the Heparin column due to a higher salt concentration in the binding buffer (250 mM NaCl). On the other hand, the binding of residual bacterial contaminants and truncated degradation products was disfavored under these salt conditions (Appendix A). The two affinity tags flanking our protein of interest were simultaneously cleaved via TEV protease digestion applied overnight at 4 °C. In the following morning, additional TEV protease was added to the dialyzed sample and the protein mixture was incubated on ice for another hour before reverse HisTrap chromatographic separation. The additional TEV protease ensured a maximal cleavage of pSUMO-Tau-NTMT, and hence a higher recovery of the cleaved full-length protein in the flow-through after the reverse HisTrap chromatography step (Figure 5). The 6xHis-SUMO-tag and TEV protease were retained on the HisTrap column (TEV protease also has an N-terminal 6xHis-tag), thereby facilitating their separation from our cleaved protein of interest. Reverse Heparin chromatography served as a final polishing step, also concentrating purified Tau-NTMT due to its ability to bind to the Heparin column, just like purified Tau-441 (Figure 5).

The final product, which eluted in the first peak of the reverse Heparin purification step, appeared to be an amalgamation of protein bands that were conjoined throughout the entire purification process (Figure 5). Our suspicion was that purified Tau tends to dimerize with both fully cleaved and partially cleaved Tau species (especially Tau with the DBD affinity tag still fused at the C-terminus). MS analysis confirmed our suspicion highlighting that the dominant gel band at 55 kilo Daltons (kDa) was pure Tau-NTMT with no affinity tags (Appendix A), whereas the bands just above it—indicated with a red arrow on the gel in Figure 5—were Tau-NTMT-DBD with no 6xHis-SUMO affinity tag (Appendix A). The high-molecular weight (HMW) bands—indicated at the top of the gel in Figure 5—were confirmed to be oligomers of pure Tau-NTMT by MS (Appendix A). It is important to note that the purity of the final product could be drastically improved by addressing incomplete TEV digestion and dimerization/oligomerization of purified Tau monomers. This adjustment was made in the purification of Tau-MTBR (a more oligomerizing and aggregation-prone domain of Tau-441) and yielded satisfactory results, as described in the next section.

#### 3.3.3. Purification of pSUMO-Tau-MTBR

As a neuron-specific microtubule-associated protein, the primary functions of Tau include regulating microtubule dynamics, maintaining neuronal cytoskeletal integrity and facilitating both anterograde and retrograde axonal transport [60,61,62]. Tau’s interaction with microtubules is mediated by its microtubule-binding domain (Tau-MTBR; residues 225–372 of Tau-441), which houses four pseudo-repeat sequences that bind to polymerized microtubule bundles with high affinity [63]. This mode of interaction allows the flexible N-terminal half to project away from bound microtubules and mediate microtubule spacing [64]. On the other hand, Tau-MTBR is usually at the center of both amyloid and amorphous aggregates in a range of diseases termed Tauopathies. The borders of the second and third pseudo-repeats house two highly hydrophobic hexapeptides, which form the core of Tau amyloids observed in patients with AD [65,66]. In other Tauopathies such as FTD, Tau forms amorphous aggregates causing neurodegeneration and dementia in affected individuals [67]. As a domain, Tau-MTBR does not phase separate on its own, rather it forms complex coacervates with polyanions such as RNA and heparin at favorable molar ratios [68].

The purification of pSUMO-Tau-MTBR presents a specific challenge, due to its known intrinsic aggregation propensity. In our hands, a solution of Tau-MTBR turned turbid in the dialysis bag upon incubation with TEV protease to cleave off the flanking affinity tags (including SUMO, which played a solubilizing role). This was indicative of LLPS among Tau-MTBR molecules with possibly nucleic acids, originally from the bacterial lysate, via charge interactions [58]. We found that performing the TEV cleavage in a buffer of higher salt (250 mM NaCl) and reducing agent (5 mM TCEP) concentrations prevented the protein solution from turning turbid. Coincidentally, like pSUMO-Tau-NTMT purification, high salt inhibited non-specific binding of bacterial contaminants to the HisTrap column and favored binding of our protein to the Heparin column rather than the 6xHis-SUMO-containing truncated fragments (Appendix A). The purification strategy where Heparin preceded HisTrap chromatographic separation still did not perform any better with this construct (data not shown), and for the same reason the approach fell short for pSUMO-Tau-NTMT. The slow expression of Tau-MTBR (16 °C, Overnight) facilitated its recovery into the soluble fraction of the bacterial cell lysate. This ensured more pSUMO-Tau-MTBR starting material was available in the lysate supernatant for the first purification step, fully saturating the HisTrap-HP-5 mL column with just one liter of bacterial lysate (Figure 6).

To improve on the purity of the final product, extra TEV protease (0.2 mg/mL) was added to the Heparin pool fractions prior to overnight dialysis and extra reducing agent (10 mM TCEP) was added to the dialyzed flow-through sample from the reverse HisTrap prior to the final polishing step (Reverse Heparin). Extra TEV protease ensured the depletion of partially cleaved pSUMO-Tau-MTBR species before taking the protein solution through Reverse HisTrap and Heparin purification steps. The efficiency of TEV proteolytic cleavage can be appreciated in the digested sample, where the large majority of full-length pSUMO-Tau-MTBR was successfully cleaved (Figure 6). However, even with extra TEV protease, there was still a population of partially cleaved Tau-MTBR with a DBD-affinity tag at their C-terminus, which was confirmed by MS (Appendix A). The addition of more TCEP to the protein solution before reverse Heparin chromatographic separation prevented oligomerization and ensured that the undesired truncations did not dimerize and elute with fully cleaved Tau-MTBR monomers, as it happened during the purification of Tau-NTMT and AF1. The complete separation of these two populations with extra TCEP resulted in elution of DBD-containing fragments together with free DBD-tag molecules in the second peak of Reverse Heparin purification step (Figure 6). Just as with Tau-441 and Tau-NTMT, fully cleaved Tau-MTBR also bound to the Heparin column, concentrating the final product in the process. Pure Tau-MTBR eluted in the first peak of the Reverse Heparin purification step, confirmed by MS analysis (Figure 6 and Appendix A).

## 4. Discussion

Recombinant expression and purification of IDPs is a field where improvements are still needed. These proteins are often challenging to purify because of their unique characteristics. However, due to their crucial roles in many cellular processes and in LLPS, they are subject of immense interest, making their production necessary in high quantity and purity. Here, we have outlined a method to overcome one of the main hurdles in their preparation, i.e., truncated degradation contaminants. The combination of two affinity chromatography steps by using a tandem-tag bacterial expression system (pSUMO) enables their easy separation from degradation products. Moreover, the two chromatographic systems applied in the method (HisTrap and Heparin) are easy to use and commercially available from a variety of sources. The tags can be removed in one step and separation of the cleaved products from the final product, while polishing the sample in the process, happens at the same time without the need of size-exclusion chromatography (SEC). By eliminating the usual polishing SEC step, the method has a fast workflow that is crucial for IDPs and suitable for industrial scale-up, thereby widening the application scope.

The robustness and versatility of the method was shown through approaching challenging and important IDP examples. The presented examples have a vast pI range, from acidic (AF1) to basic (Tau-441, Tau-NTMT, Tau-MTBR). As a matter of fact, the Tau domain constructs have an increasing pI as the length decreases, yet the method still performs well independently of charge characteristics. When compared to the existing purifications in the literature, our method had a clear advantage in the examples shown. For instance, in case of AF1, the purity of the sample was substantially increased, with degradation contaminants falling almost below detection limit: only a small amount of partially cleaved protein could be detected due to the formation of mixed dimers between partially cleaved and fully cleaved AF1. This can be overcome by adding more reducing agent before the reverse Heparin purification step, as demonstrated in the case of Tau-MTBR. In addition to improving purity, we also achieved significantly increased yields. In the case of Tau and its domain constructs, the purity of the sample was as good or better than that reported in the literature, with practically undetectable degradation contaminants [69,70]. Our method also allows the elimination of the boiling step that is favored in the IDP field, but which can have adverse effects on sample quality (causing oxidation, deamidation, etc.), thereby compromising downstream experiments. In the case of Tau constructs, the yield was comparable or slightly better than that obtained with existing purification methods [69,70].

Another level of versatility of the method is changing the order of the affinity chromatography steps. By starting with HisTrap followed by Heparin, the method seems to be more general. However, starting with Heparin followed by HisTrap has the advantage of enabling the application of an increased amount of reducing agent during cell lysis. Furthermore, the elution buffer of the Heparin chromatography step is more compatible with the loading buffer of the HisTrap purification step, hence requiring less buffer exchange steps to ensure a faster workflow. Nonetheless, our experience with this system has shown us that it is best to optimize which order works best for a given target protein.

The findings in this study demonstrate that novel avenues of IDP purification are conceivable, overcoming the existing hurdles to ensure IDP products of high quantity and purity. To emphasize the generality of our method, we would like to produce labeled proteins for NMR experiments as a possible future application. For ongoing projects in our laboratory, we plan to purify labeled Tau-MTBR, alongside a few other IDPs, by making use of the presented method.

## Figures and Tables

**Figure 1 biomolecules-12-01566-f001:**
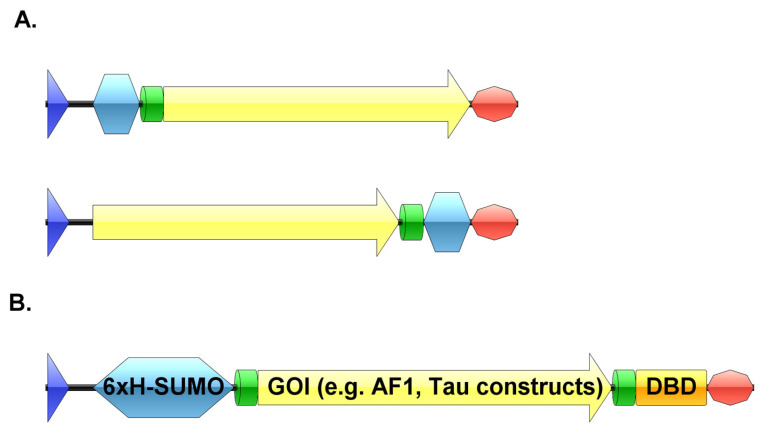
(**A**) Schematic representations of expression plasmids with either an N-terminal or a C-terminal affinity tag (blue—promoter region, silver blue—affinity tag, green—proteolytic cleavage site, yellow—gene of interest, orange—second affinity tag, red—terminator sequence). (**B**) Schematic representation of the pSUMO plasmid with tandem tags (blue—promoter region, silver blue—6xHis-SUMO-tag, green—TEV cleavage site, yellow—gene of interest, orange—DBD, red—terminator sequence). Created by IBS software [50].

**Figure 2 biomolecules-12-01566-f002:**
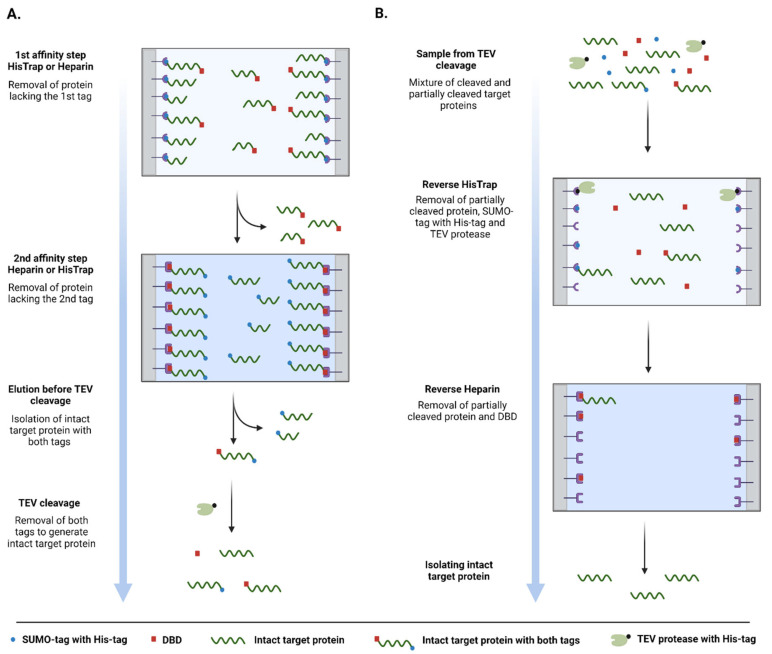
Schematic representation of the workflow of the tandem-tag purification method. (**A**) Representation the first two affinity steps before TEV cleavage. (**B**) Representation the reverse affinity steps after TEV cleavage. (Figure was created with BioRender.com).

**Figure 3 biomolecules-12-01566-f003:**
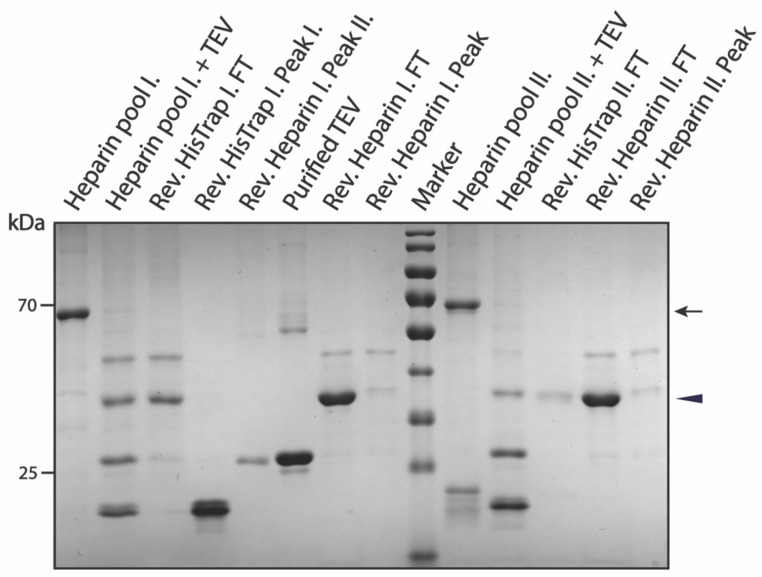
SDS-PAGE summarizing pSUMO-AF1 purification. The black arrow indicates intact AF1 having both affinity tags, while the black triangle indicates the cleaved final product. The lane ‘Purified TEV’ on the gel was loaded with a sample of TEV protease enzyme that was purified in-house and used in affinity-tag cleavage of all purified protein samples presented in this paper.

**Figure 4 biomolecules-12-01566-f004:**
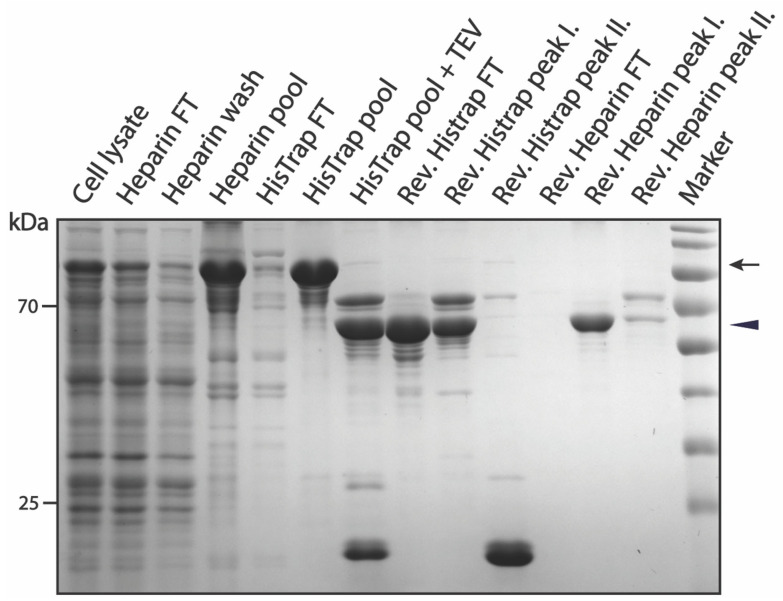
SDS-PAGE summarizing pSUMO-Tau-441 purification. The black arrow indicates intact Tau-441 having both affinity tags, while the black triangle indicates the cleaved final product.

**Figure 5 biomolecules-12-01566-f005:**
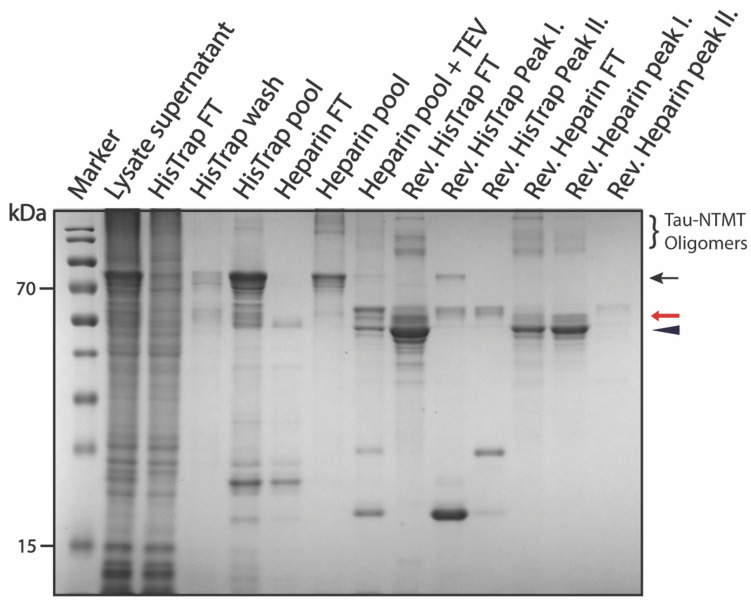
SDS-PAGE summarizing pSUMO-Tau-NTMT purification. The black arrow indicates intact Tau-NTMT with both affinity tags, the red arrow indicates Tau-NTMT with a DBD-affinity tag at its C-terminus and the black triangle indicates the fully cleaved Tau-NTMT product.

**Figure 6 biomolecules-12-01566-f006:**
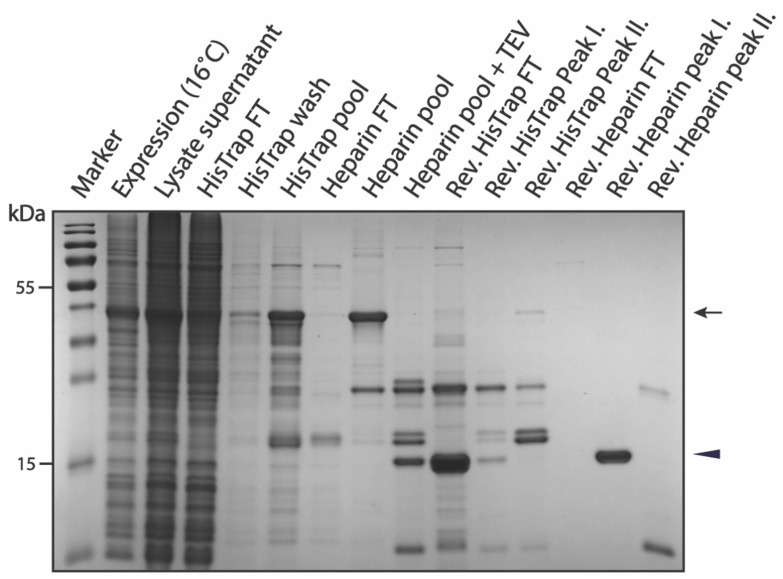
SDS-PAGE summarizing pSUMO-Tau-MTBR purification. The black arrow indicates intact Tau-MTBR having both affinity tags, while the black triangle indicates the cleaved final product.

## Data Availability

The data presented in this study are available upon request from the corresponding author.

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
