# Peer review of "A Novel Tandem-Tag Purification Strategy for Challenging Disordered Proteins"

_biomolecules, 2022, doi:10.3390/biom12111566_

Round 1

Reviewer 1 Report

Please, find the enclosed document.

Author Response

Dear Reviewer,

First and foremost, we would like to thank you for considering our manuscript for publication and we appreciate the suggestions to improve our article in all aspects.

In the re-submitted version of our manuscript, we addressed all points raised, and we believe the edited chapters of our article have visible improvements.

Please find our point-by-point replies and changes in specific in the attached document.

Reviewer 2 Report

The authors report an efficient purification strategy of intrinsically disordered protein (IDP) expressed in E. coli cells. The authors demonstrate that IDPs, some of them are prone to oligomerize and phase-separate into liquid droplet, can be prepared with high purity at milligram scale, even without gel-filtration purification. One of the advantages of the proposed strategy is that the strategy enables fast purification and even larger scale purification. Furthermore, it is worth to emphasize that the strategy has been shown to be efficient to purify the proteins responsible for liquid-liquid phase separation (LLPS) that is generally difficult to handle due to its high tendency of oligomerization and  promiscuous binding to other proteins and nucleic acids from the host cells. Given the increasing demands for production of LLPS proteins for structural, biochemical, and biophysical studies aiming to elucidate the molecular mechanisms of LLPS-related events, this strategy will attract interests from many readers of Biomolecules. The reviewer acknowledges importance of the manuscript, but found several points needed to be considered and refined in prior to the publication. 

Major points:

-       The authors propose new strategy of tandem-tagging for efficient expression and purification of the protein, but there is very limited information about the previous studies regarding double-tagging in the introduction section. The idea of tandem-tagging has been proposed and demonstrated in decades ago (PMID: 2543972) and several variations have been proposed so far (e.g. PMID: 33433908; PMID: 29997597). To highlight the significance and novelty of the strategy, the authors need to explain the related previous works. Although a couple of works (refs 40, 41) are described in the result section, but these information should be first provided in the introduction section.

Specific points

-       Page 16, line 570: Although the authors claim “the yield was comparable or a slightly better than with the existing methods”, the evidence is unclear. The authors need to specify the data or cite previous study to be compared with their tandem-tag method. 

-       Page 16, line 566: “the literature” needs to be specified with citation.

-       Page 3, line 112: An abbreviation “NTMT” and “MTBR” needs to be clarified. If these indicate a specific region of Tau, residue boundaries need to be specified. The same thing applies to “Tau-441”  

-       Page 5, line 207: “one litre of bacterial pellet” seems to be corrected as “bacterial pellet from one litre of the medium”.

-       Figure 3: It is helpful if the authors add the labeling to clarify which lanes are for the samples from His-trap pool I, and which lanes are from II. There are two lanes labeled with “Rev. Histrap FT”.

-       Page 12, line 471: It would be helpful if the bands corresponding to “the bands just above it” are indicated in Figure 5.

-       Page 16, line 585: SDS-PAGE

-       Page 4, line 165, Page 5, line 203 and 230: “Terrific broth (TB)” can be corrected to “TB”, since the abbreviation is first described in page 3, line 133.   

Author Response

Dear Reviewer,

First and foremost, we would like to thank you for considering our manuscript for publication and we appreciate the suggestions to improve our article in all aspects.

In the re-submitted version of our manuscript, we addressed all points raised, and we believe the edited chapters of our article have visible improvements.

Please find below our point-by-point replies and changes in specific.

Reviewer 3 Report

1.     The authors should check the manuscript carefully, there are several grammatical mistakes throughout the manuscript.

2.     The author should check the usage of comma and period throughout the manuscript example: Line 83.

3.     They should check the formatting for the citations, example: line 64, it should be [11,25-28], There are several errors in citation

4.     Line 142, which DNase has been used here?

5.     Line 183: it should be SDS-PAGE

6.     Line 328 and 329 is not required

7.     In Figure 1, the author can include actual diagram of any one plasmid they generated for this work

8.     The author should improve figure 2. It is difficult for the reader to understand the purification strategy from the figure

9.     Line 374: what do they mean by chapter 3.3.3?

10.  Did the author compare their method with the other published method of purification of IDR like boiling?

11.  In figure 4, the author should represent the gel in similar fashion as in other figures. (left to right: cell lysate to Rev heparin peak II)

Author Response

(The authors gave the same response as above.)
